# Considerations for Newborn Screening for Critical Congenital Heart Disease in Low- and Middle-Income Countries

**DOI:** 10.3390/ijns6020049

**Published:** 2020-06-14

**Authors:** Bistra Zheleva, Sreehari M. Nair, Adriana Dobrzycka, Annamarie Saarinen

**Affiliations:** 1Children’s HeartLink, Minneapolis, MN 55436, USA; adriana@childrensheartlink.org; 2National Health Mission, Kerala, Thiruvananthapuram 695 035, India; drsreeharim@gmail.com; 3Newborn Foundation, Minneapolis, MN 55116, USA; annamarie@newbornfoundation.org

**Keywords:** congenital heart disease, critical congenital heart disease, pulse oximetry, congenital heart surgery, population health

## Abstract

We propose several considerations for implementation of critical congenital heart disease (CCHD) screening for low- and middle-income countries to assess health system readiness for countries that may not have all the downstream capacity needed for treatment of CCHD. The recommendations include: (1) assessment of secondary and tertiary level CHD health services, (2) assessment of birth delivery center processes and staff training needs, (3) data collection on implementation and quality surgical outcomes, (4) budgetary consideration, and (5) consideration of the CCHD screening service as part of the overall patient care continuum.

## 1. Introduction

Newborn screening for critical congenital heart disease (CCHD) using pulse oximetry has gained momentum in recent years through the research and advocacy of committed parents, clinicians and public health leaders. Today, it has been added to uniform screening panels in many countries and it is considered one of the important ways to find children with CCHD at the earliest possible time to ensure that babies have a chance of survival through early surgical intervention.

It has been exciting to see how this has developed and grown into a global movement, from the early days when it started with just a few champions in several high-income countries. We have also seen that many low- and middle-income countries (LMICs) have started considering adding this screening to their universal screening guidelines. It is very likely that this has been driven by the climbing rank of CHD as a contributor to infant mortality. The Global Burden of Disease study showed us that from 1990 to 2017 the rank of CHD increased from ninth to seventh globally and from 17th to 11th in low-socio-demographic index countries, and improvement in CHD mortality lagged behind that of other causes. Of the 261,247 deaths caused by CHD globally in 2017, 180,624 were among infants and most deaths occurred in countries in the low and low-middle SDI quintiles [1]. As countries strive to achieve the 2030 Sustainable Development Goals agreed upon in 2015 [2], many, especially middle-income and upper-middle-income countries, have realized that addressing mortality from CHD will help with driving down overall infant mortality (IMR) and under-five mortality rates (U5MR). Many governments have set their own national and subnational goals to achieve these reductions, which tells us that newborn screening for CCHD can play a critical role in this process, while also supporting sweeping initiatives around equity and access to care for noncommunicable diseases globally. In 2013, Zühlke and Vaidyanathan [3] recommended to look at newborn pulse oximetry screening feasibility based on IMR and to look closely at IMR causes, concluding that countries with lower IMR may be better suited for the implementation of such screening services and embedding them into already existing newborn screening programs.

## 2. CCHD Screening in LMICs

Just like with any new technology-driven health intervention, implementers have to be mindful of the context into which the intervention will be implemented and the challenges that could be present. While pulse oximetry is simple, quite inexpensive, and relatively easy to implement, it does require downstream capacity in a well-functioning secondary and tertiary care structure for the screening to be successful. The screening process is just one step in a lifelong care continuum for patients with congenital heart disease (Figure 1). Quick access to secondary and tertiary healthcare services is critical for the successful diagnosis, confirmation and subsequent referral of newborns to tertiary care centers for treatment, including surgery. Requirements for developing successful pediatric cardiac in-hospital services include robust infrastructure, quality equipment, a high level of skill among caregivers, cohesive multidisciplinary teams, and supportive hospital administration. Requirements for secondary services supportive of successful newborn screening include the ability to confirm the CCHD diagnosis at the level of either the delivery hospital or district (referral) hospital and availability of a safe and timely newborn transport system. Appropriate financing for scaling the services is imperative too, as financial protection for patients’ families is critical given the surgical intervention and the potentially ongoing costs of follow-up care for CHD.

In another study in 2016 [4], Van Niekerk and colleagues tested pulse oximetry screening in South Africa and, while they found it was feasible, the staffing shortages made it difficult to implement. They found that, while it was supported and accepted by staff and parents, nursing staff shortages led to workflow problems and prevented its routine application before discharge, which in turn led to more errors in the test.

In our experience in Kerala, the government took a population health approach to managing CHD and introduced pulse oximetry CCHD newborn screening as part of the overall CHD care continuum. While most of the downstream categories were developed or somewhat developed in Kerala, we also found that the process of training the healthcare workforce was not always straightforward. It took some time to perfect the use of the technology, and an additional challenge was the absence of a good comparison with global guidelines or national guidelines on pairing pulse oximetry with physical examination, including heart sound auscultation, an approach successfully tested in China in 2014 and 2017 [5,6]. In the United States, physical examination is not a mandatory part of the screening process because, as a general rule, most births happen in a delivery facility and, prior to discharge, as part of routine newborn care, all newborns are seen by a pediatrician trained to listen for heart problems. LMICs, on the other hand, rarely offer this service. In the case of Kerala, the Indian state with the best health outcomes, part of the physical exam has to be done by delivery center nurses and requires additional training in it. While a protocol was eventually developed for nurses to do pulse oximetry, combining it with a physical exam specifically looking at respiratory rate, pulse/heart rate, color of the baby for any cyanosis, and heart sounds, including any murmurs and peripheral pulses, this setback was indicative of what the situation could look like in other, less-well-resourced settings.

So far, CCHD newborn screening has been widely implemented and guidance has been released by several high-income countries, which has been modified to fit the needs of LMICs piloting or implementing screening, with varying degrees of success, but not much has been done at a global level. A number of LMICs have started implementation, with China notably doing the largest studies in the world, and ultimately adding a “dual index” method of newborn CCHD screening as part of a national implementation program in 2018. Efforts are under way in approximately 66 countries currently piloting or implementing newborn screening. It is important to note that pulse oximetry screening among newborns has also proven to be an effective tool for detecting otherwise asymptomatic cases of pneumonia, sepsis, PPHN, non-critical CHD and other conditions associated with hypoxemia. As such, newborn screening can help health workers in identifying conditions for which follow up testing and treatments are available and may allow families of children with less critical cardiac conditions to plan for care and essential medical services in the future. With these efforts, countries are hoping to save many lives and address some of their national infant mortality rates.

## 3. Recommendations for CCHD Screening in LMICs

In light of these experiences, we wanted to offer several recommendations for future implementation of CHD newborn screening using pulse oximetry in LMICs.
Without ready access to definitive diagnosis, safe transportation and high-quality surgical and interventional cardiology services, newborn screening cannot save as many lives as it should. It will help detect cases, but, unless these babies can access surgical services, many will not survive or will live a life with serious disability. It is critical for policy implementation to include an assessment of the existing infrastructure to assure effective utilization of all available resources, rather than considering making investments in new infrastructure. This would involve holding consultations with a broad representation of the whole health sector, including public and private healthcare providers, patients and families, policymakers and the biomedical industry.Prior to implementation, it is necessary to thoroughly investigate what the process of birth delivery in the country is and who would be best positioned to administer the test. This includes developing a good understanding of what the healthcare workers administering the test’s usual scope of practice is and what their upskilling and training needs would be.It is critical to collect data on the new process and to study it as it is being implemented, as well as on the quality of the surgical outcomes of the children detected through pulse oximetry screening. One of the biggest influences of newborn CCHD screening is that it finds children who were invisible before, providing a more accurate picture of the burden of disease, and this can create a strong impetus for downstream capacity development and infrastructure investments, which will ultimately benefit many more children with CHD. In the case of Kerala, the government developed a state-wide registration system, which helped with the subsequent data analysis.The implementing parties need to back their plans with robust budgetary considerations, as this is not a simple clinical practice change but rather a health policy decision affecting lifelong care for children with CCHD. We have found that this implementation is more successful when it is led by the government, given that health policy officials have the ability and experience mandating screenings for other diseases in the newborn period.Pulse oximetry CCHD screening in newborns should also ideally be considered as one of the steps to screen for CHD, given that there is no one method that can detect it all. A population health approach to CHD screening should also include antenatal screening and follow up screening at different points in the first several years of life, taking advantage of the frequent primary care interactions for regular vaccinations, for example. This recommendation is especially important for LMICs, where there are still significant numbers of undetected children with CHD. It is also critical to look at where CHD care is needed in the different levels of the healthcare system (Figure 2) [7].

To conclude, newborn screening for CCHD provides an opportunity to change the status of heart disease in children from an invisible disease to one integral to eliminating preventable child deaths. The recommendations we present would make this a policy decision rooted in data and evidence that will avoid the disheartening tendency of some policymakers to wait with development of plans for CHD because of its perceived high costs. We enthusiastically support countries introducing and mandating it but want to offer caution that readiness for this important test has to be considered prior to its implementation and that it needs to be part of a comprehensive plan for the overall development of pediatric cardiac services.

## Figures and Tables

**Figure 1 IJNS-06-00049-f001:**
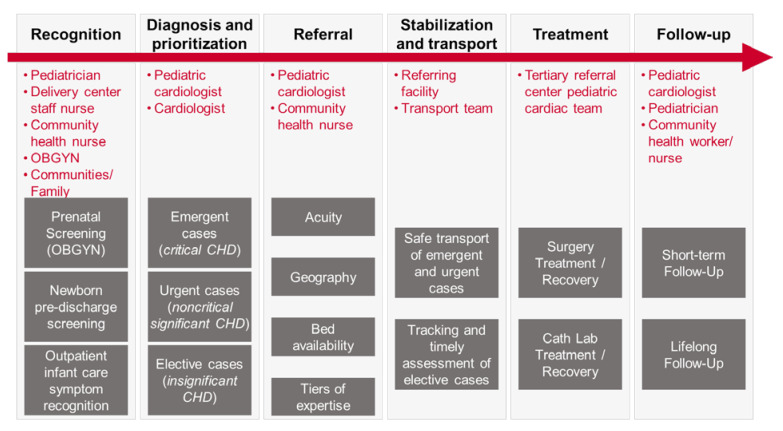
Congenital heart disease (CHD) patient care continuum.

**Figure 2 IJNS-06-00049-f002:**
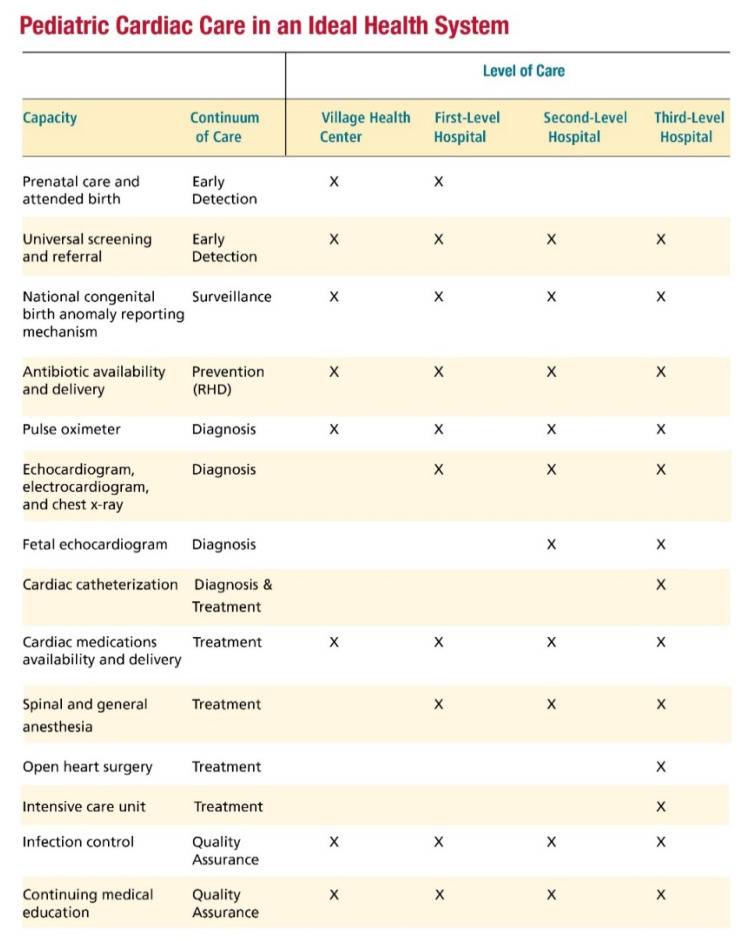
Pediatric cardiac care in an ideal health system. (X denotes service needed at that level).

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
