# Peer review of "Considerations for Newborn Screening for Critical Congenital Heart Disease in Low- and Middle-Income Countries"

_2409-515X, 2020, doi:10.3390/ijns6020049_

Round 1

Reviewer 1 Report

Interesting and well written guidelines for newborn screening for CHDs using pulse oximetry, which is easy to implement in low to medium income countries.  

I was wondering if genetic tests should not be included in the follow up and diagnosis.

Minor comment: Line 110, sentence is ackward and there seems to be a typo: "interventional cardiology services, newborn screening cannot not save as many lives as it should".

Author Response

Thank you for the review and for the comments. Here are our responses.

  1. We believe the genetic testing suggestion refers to the care continuum. Similar to newborn screening, genetic testing would be recommended with caution, because of social, ethical and potentially cultural barriers to its efficient implementation in LMICs, especially poor financial accessibility of the services and inadequate medical training in clinical genetics. This was confirmed by a 2018 systematic review (https://www.nature.com/articles/s41436-018-0090-9) and by 2011 WHO consultation (https://apps.who.int/iris/bitstream/handle/10665/44532/9789241501149_eng.pdf?sequence=1&isAllowed=y). For that reason, we have decided not to include it. We are also well aware that there is no consistent genetic screen in the US too, and it's done for some and not other CHDs.
  2. Changed the sentence on line 110 to “Without ready access to definitive diagnosis, safe transportation and high-quality surgical and interventional cardiology services, newborn screening cannot not save as many lives as it should.”

Reviewer 2 Report

The authors propose and elaborate several considerations for implentation of screennig for critical congenital heart disease (cCHD) in low- and middle-income countries. The manuscript is clear and well written. Well known and generally accepted principles of screening are applied to the field of cCHD, thereby providing a usefull basis for discussion when implenting a new screening method is considered in the above mentioned context.

Author Response

Thank you so much for the comments. We understand that no changes are necessary to the manuscript.